# The importance of using a multi-dimensional scale to capture the various impacts of precarious employment on health: Results from a national survey of Chilean workers

Alejandra Vives [1,2]*, Tarik Benmarhnia [3], Francisca González [4], Joan Benach [5,6,7]

1 Department of Public Health, School of Medicine, Pontificia Universidad Católica de Chile, Santiago, Chile, 2 Center for Sustainable Urban Development (CEDEUS), Conicyt/Fondap, Santiago, Chile, 3 Department of Family Medicine and Public Health & Scripps Institution of Oceanography University of California, San Diego, CA, United States of America, 4 Department of Mathematics, Universidad Técnica Federico Santa María, Valparaíso, Chile, 5 Department of Political and Social Sciences, Health Inequalities Research Group, (GREDS-EMCONET), Universitat Pompeu Fabra, Barcelona, Spain, 6 Johns Hopkins University-Pompeu Fabra University Public Policy Center, Barcelona, Spain, 7 Transdisciplinary Research Group on Socioecological Transitions (GinTRANS2), Universidad Autónoma de Madrid, Madrid, Spain

* alejandra.vives@uc.cl

**Data Availability Statement:** The data underlying the results presented in the study are available from http://epi.minsal.cl/encuesta-enets/.

## Abstract

### Background

Social epidemiologic research in relation to the health impacts of precarious employment has grown markedly during the past decade. While the multidimensional nature of precarious employment has long been acknowledged theoretically, empirical studies have mostly focused on one-dimensional approach only (based either on employment temporariness or perceived job insecurity). This study compares the use of a multidimensional employment precariousness scale (EPRES) with traditional one-dimensional approaches in relation to distinct health outcomes and across various socio-demographic characteristics.

### Methods

We used a subsample of formal salaried workers (n = 3521) from the first Chilean employment and working conditions survey (2009–2010). Multilevel modified Poisson regressions with fixed effects (individuals nested within regions) and survey weights were conducted to estimate the association between general health, mental health and occupational injuries and distinct precarious employment exposures (temporary employment, perceived job insecurity, and the multidimensional EPRES scale). We assessed the presence of effect measure modification according to sex, age, educational level, and occupational class (manual/non-manual).

### Results

Compared to one-dimensional approaches to precarious employment, the multidimensional EPRES scale captured a larger picture of potential health effects and differences across

**Funding:** This work was supported by CONICYT/ FONDECYT [1171105] to AV; and partially supported by Concurso Profesores Extranjeros Convocatoria 2016, Pontificia Universidad Católica de Chile [PVE16007] to AV. The funders had no role in study design, data collection and analysis, decision to publish, or preparation of the manuscript.

**Competing interests:** The authors have declared that no competing interests exist.

subgroups of workers. Patterns of effect measure that modification were consistent with the expectations that groups in greater disadvantage (women, older individuals, less educated and manual workers) were more vulnerable to poor employment conditions.

## Conclusions

Multidimensional measures of precarious employment better capture its association with a breath of health outcomes, being necessary tools for research in order to strengthen the evidence base for policy making in the protection of workers' health.

## Introduction

The flexibilization of employment relationships of the past decades has led to the growth of precarious employment [1]. Important numbers of workers are affected by this precarisation of employment, motivating the study of its potential effects on the health and wellbeing of workers and their families. Such research has grown markedly during the past decades, but while scholars from different disciplines have long acknowledged that there are several dimensions to precarious employment, the main approaches to produce epidemiologic evidence have been largely one-dimensional, focusing primarily on job instability. In this context, the development of clearer, more comprehensive definitions of the concept of precarious employment and its operationalization are among the main research gaps that have been identified in the advancement of the precarious employment and health research agenda [2]. This paper aims at highlighting the importance of using a multidimensional theory-based employment precariousness scale in comparison with traditional one-dimensional empirical approaches, not only to improve or understanding of the problem but also to enhance the contribution that epidemiologic evidence can make to inform policy making towards the improvement and protection of workers' health.

The two main one-dimensional approaches, which have contributed significantly to the available epidemiological evidence in the last decades, can be grouped into perceived job insecurity studies and temporary employment studies. Perceived job insecurity is the subjectively perceived likelihood of involuntary job loss [3], generally measured as the overall concern regarding the continuity of the job in the future [4]. Research on job insecurity shows there to be effects upon several health outcomes, the most studied of which are its effects on mental health [5]. However, perceptions of job insecurity may be elicited by different contextual determinants, including events in the private life of individuals, such as the appearance of health problems. Also, the magnitude of job insecurity in the face of external threats to the continuity of the job may vary considerably between individuals due to personal attributes. It is thus a largely "private" experience, more closely linked to individual psychology than to the actual employment relationship [6], but provides only a partial picture of precarious employment and how it may affect health [7]. The second one-dimensional approach, developed partly in response to the limitations of job insecurity studies, addresses the health effects of different types of 'temporary employment' jobs by comparing them to permanent jobs, which often are considered the "ideal" standard of employment, secure and non-precarious [8]. Temporary workers have been found to be exposed to worse working conditions and harmful exposures, and to be consistently associated with worse mental health and more workplace injuries [9]. Despite this, studies have produced contradictory findings, whereby some have found an inverse association between type of contract and health, or no association at all [10]. This may

be partly explained because not all temporary jobs are necessarily precarious, but mainly because the increased utilization of temporary employment has expanded this precarisation to all contract types, such that many permanent jobs are also, to some extent, precarious. This implies that the one-dimensional temporary employment approach may produce exposure misclassification and ultimately underestimate the impact of precarious employment on health.

The limitations of one dimensional approaches to precarious employment have motivated the advancement of instruments that can account for its multidimensional nature. Several proposals for a multidimensional conceptualization of employment precariousness have emerged, but they have seldom been operationalized to use in epidemiologic research. One exception is the Employment Precariousness Scale (EPRES) developed by the GREDS-EMCONET research group drawing on Rodger's multidimensional definition [11]. The EPRES encompasses not only the uncertainty of continuing employment; but also other key aspects of employment relationships, organized into six dimensions: 'temporariness' or employment instability; 'disempowerment' (individualized vs collective bargaining); 'vulnerability' (worker defenselessness to unacceptable workplace practices); 'wages' (low or insufficient; possible material deprivation); 'rights' (entitlement to social security benefits); and 'exercise rights' (powerlessness, in practice, to exercise workplace rights) [11, 12]. Recent research has described associations between the EPRES score and mental and general health [13, 14], as have studies using proxy multidimensional measures with European data from the European Working Conditions Surveys.

Nonetheless, to our knowledge no epidemiological effort has compared yet the health impacts of multidimensional and one-dimensional approaches to precarious employment in order to assess to which extent they differ in their association with health. It is also important to conduct such comparison across different outcomes that include mental health, physical health and occupational health to detect the potential different mechanisms involved.

In parallel, while most studies describing employment conditions tend to coincide in finding that groups in labour market disadvantage are more frequently in precarious employment, few studies have compared its health effects across sub-groups of workers, except for some studies stratifying by gender and, in few cases, occupation [15]. Assessing heterogeneity in the impact of one-and multi-dimensional measures is particularly important to identify vulnerable subgroups, so as to shape targeted interventions.

In Latin America epidemiological research on formal precarious employment and health is scant, in part due to the greater attention informal employment convenes given its pervasive presence in the region´s labour markets. In Chile, while informal employment still occupies a significant portion of the labour force, the majority of jobs are formal salaried jobs, but affected by significant instability [16]. The full version of the EPRES was included in the first Chilean employment and working conditions survey (ENETS), together with the main one-dimensional measures of precarious employment [17]. This provides a unique opportunity for comparing measures of precarious employment across different health outcomes. Initial descriptive analyses showed a high proportion of workers perceiving their jobs as insecure and suggest there to be a higher prevalence of poor health among workers in more insecure or precarious jobs [18].

Hence, using data from the first Chilean employment and working conditions survey (ENETS), which offers high quality data on employment and worker health, this study compares the association of one-dimensional measures (temporary employment, perceived job insecurity) and the multidimensional EPRES scale with general self-perceived health and mental health, the most studied outcomes of precarious employment, as well as self-reported occupational injuries, an objective occupational health outcome.

As a secondary objective, this study aims to identify heterogeneity across sub-groups to identify those especially vulnerable to poor employment conditions in one or another measure, by examining whether the observed associations are modified by four of the main axes of labour market inequality: gender, age, educational attainment and occupation (manual or non-manual).

## Materials and methods

This study has been approved by Pontificia Universidad Católica´s School of Medicine Institutional Review Board (IRB), approval number 12–128. Consent was not required since secondary data were used and analyzed anonymously.

### Data

The study sample comes from the first Chilean survey on Employment conditions, Work, Health and Quality of life (ENETS) conducted in 2009–2010, and is representative of the national workforce at national, regional and urban and rural levels [17]. Sample selection followed a multistage, stratified random sampling procedure, with an overall response rate was 73.9%. Interviews were conducted in the participant's household by trained interviewers; participation was voluntary and confidential. Completely anonymized data sets are directly available from the Ministry of health's webpage.

Because the EPRES is specifically devised for formally employed workers, the study sample will be restricted to workers in salaried employment with a formal work contract (n = 3521), thus making proper use of the scale and results comparable internationally.

### Study variables

**Health outcomes.**   Self-reported general health was assessed by the single item "In general would you say your health is. . ." with a dichotomous outcome variable (1: fair, less than fair, bad, very bad; 0: more than fair, good, very good).

Mental health was measured with the 12-item version of the General Health Questionnaire (GHQ-12) for non-specific psychiatric morbidity. The GHQ Likert scoring method was used to assess the magnitude of psychological distress [19], classifying subjects into poor mental health if they belong to the 4th quartile of the distribution.

Occupational injuries (yes/no) were all self-reported non-fatal workplace injuries in the 12 months prior to interview [20].

**Precarious employment variables.**   Exposure was measured with two one-dimensional and one multidimensional approach: i) perceived job insecurity, measured as the concern about being fired or not having the contract renewed (Never and rarely = not insecure; almost always and always = insecure), ii) temporary employment, including both fixed-term and non-fixed term temporary contracts, compared to permanent jobs, and iii) the Employment precariousness scale (EPRES-Ch) for the multidimensional assessment, which encompasses 6 dimensions: 'temporariness' (3 items), 'disempowerment' (3 items), 'vulnerability' (5 items), 'wages' (3 items), 'rights' (3 items), and 'capacity to exercise rights' (5 items). EPRES subscale scores are computed as a simple average transformed into a 0–4 scale and then averaged into a global EPRES score which ranges from 0 (not precarious) to 4 (most precarious), and which we divided into tertiles.

**Sociodemographic and occupational variables.**   The variables used were sex, age (as a continuous variable), educational attainment (primary or less, secondary, trade school, and university), urban or rural residency, occupational class based on the International Standard Classification of Occupations (ISCO-88) and grouped into manual and non-manual,

economic activity (International Standard Industrial Classification of all Economic Activities (ISIC-Rev.2, 1968). Region of residency (15 Chilean regions) was included in the multilevel models.

### Statistical analyses

Multilevel Poisson (modified with robust variance for binary outcomes) regressions with fixed effects, where each individual was nested in his region, were conducted to estimate adjusted prevalence ratios (PR) [21] representing the association between the health outcomes (n = 3) of interest and the exposure to each employment conditions (n = 4). A likelihood ratio test was used to consider the presence of within-region variability for each model. Covariates and causal pathways were defined a priori based on the literature. Analyses were thus adjusted for gender, age, and occupational class. We created tertiles for employment precariousness as an exposure of interest to investigate potential non-linear patterns. We considered the complex survey design by including the survey weights provided by ENETS in our analysis [17]. We performed complete case analysis for all our analyses.

We conducted sensitivity analyses by excluding from the models i) individuals aged more than 65 years; ii) individuals with limiting illness in the 12 months preceding the survey and ii) individuals working in the public sector.

As a secondary analysis, we assessed if the association between each of the health outcomes of interest (n = 3) and each of the precarious employment exposures (n = 3) was heterogeneous according the following variables: sex (men vs. women); age (considered as continuous variable, 1-year units); level of education (lowest vs. highest); occupational class (non-manual vs. manual). We assessed the presence of heterogeneity through the inclusion of an interaction product term in the Poisson models described above. We therefore obtained, for each model, the interaction term coefficient and its 95% confidence Interval (95% CI) to assess potential heterogeneity. When level of education was considered in the interaction term, occupational class was excluded from the models. All analyses were performed with Stata 14 SE.

## Results

The majority of the sample are men (66%), aged 25 to 44 years (51.9%), has secondary education (58.7%), urban residence (86.1%), are non-qualified workers (62.7%; 31.5% non-manual and 31.2% manual); 32.6% work in communal services and 19% in commerce (wholesale). Up to 16.5% has temporary employment and 30.5% reports job insecurity. Employment precariousness scores ranged from 0 to 3.51, (median = 1.28). The prevalence of poor general health was 21.3%, poor mental health (third tertile) concentrated 361%, and 6.3% of workers reported having suffered at least one occupational injury in the preceding 12 months. (Table 1)

We found that poor general health was associated with employment precariousness in the form of a gradient, and with job insecurity, but not with temporary employment. Poor mental health was associated with the third tertile of employment precariousness and with job insecurity, but not with temporary employment. (Table 2)

Occupational injuries were not associated with neither job insecurity nor temporary employment. Instead, a gradient association was observed with employment precariousness, reaching 2.48 (95%C.I.: 1.42–4.33) among workers in the third tertile.

Results for the heterogeneity assessments showed that women were more vulnerable than men in the association between employment precariousness and general and mental health, while men were more vulnerable than women in the association between job insecurity and type of contract with mental health.

**Table 1. Sample characteristics.**

| Variable | | % |
|---|---|---|
| Sex | Men | 64.1 |
| | Women | 35.9 |
| Age groups (years) | 15–24 | 12.9 |
| | 25–44 | 49.1 |
| | 45–64 | 35.7 |
| | 65+ | 2.3 |
| Educational attainment | Basic | 18.3 |
| | Secondary | 61.6 |
| | Trade school | 10.3 |
| | University | 9.8 |
| Zone | Urban | 89.1 |
| Occupation | Qualified non-manual | 20.9 |
| | Non-qualified non-manual | 31.5 |
| | Qualified manual | 16.3 |
| | Non-qualified manual | 31.2 |
| Economic activity | Agriculture, hunting and forestry | 12.0 |
| (ISIC Rev.4) | Mining and quarrying | 3.8 |
| | Manufacturing | 13.0 |
| | Electricity, gas and water supply | 1.3 |
| | Construction | 8.9 |
| | Wholesale, hotels & restaurants | 19.0 |
| | Transport, storage and communication | 6.4 |
| | Real estate, renting and business activities | 2.2 |
| | Other social community services | 32.6 |
| Type of contract | Temporary | 16.5 |
| Job insecurity | Yes | 30.5 |
| Employment Precariousness (EPRES) | T1 | 34.0 |
| | T2 | 29.9 |
| | T3 | 36.1 |
| Poor mental health | Yes | 16.6 |
| Poor self-reported health | Yes | 21.3 |
| Occupational injuries | Yes | 6.3 |

Salaried formal workers, Chile 2009–10. (n = 3.521).

For age, we found effect modification for all associations with general and mental health indicating that the eldest were more vulnerable. For education results only indicated a lower risk for those in higher education in the association with occupational injuries and type of contract. Finally, we found non-manual workers to be less vulnerable than manual workers in most associations (see S1 Table)

After conducting sensitivity analysis, our conclusions were not affected.

## Discussion

To our knowledge, this is the first study to analyze and compare the health-related associations of the two main one-dimensional measures of precarious employment to a multidimensional measure of precarious employment. We show that different precarious employment exposures may lead to different conclusions in terms of health associations. Our main findings are that

**Table 2. Prevalence rate ratios for the associations between study exposures and outcomes.**

| | | Poor General Health | | | Poor Mental Health | | | Occupational Injuries | | |
|---|---|---|---|---|---|---|---|---|---|---|
| | | **PRR** | **LCI** | **UCI** | **PRR** | **LCI** | **UCI** | **PRR** | **LCI** | **UCI** |
| Employment precariousness | T1* | | | | | | | | | |
| | T2 | 1.59 | 1.14 | 2.22 | 1.23 | 0.72 | 2.11 | 2.21 | 1.17 | 4.17 |
| | T3 | 3.07 | 2.20 | 4.29 | 2.38 | 1.43 | 3.96 | 2.48 | 1.42 | 4.33 |
| Job Insecurity | Not insecure* | | | | | | | | | |
| | Insecure | 1.49 | 1.14 | 1.94 | 1.92 | 1.42 | 2.60 | 1.52 | 0.88 | 2.62 |
| Type of Contract | Permanent* | | | | | | | | | |
| | Temporary | 0.99 | 0.74 | 1.33 | 1.23 | 0.86 | 1.76 | 0.99 | 0.56 | 1.76 |

Chile, salaried workers 2009–10.

PR: Prevalence ratios; LCI: Lower confidence interval; UCI: upper confidence interval.

* Reference group.

while employment precariousness and job insecurity were associated with all three health outcomes considered in this study, while temporary employment was associated with none, and that the multidimensional approach was the most sensible to both the association with health and to subgroup heterogeneity.

Against expectations, we found no associations between temporary employment and the three health outcomes studied. This finding adds to the body of contradictory or heterogeneous research findings. However, contrary to the hypothesis that this may be due to the heterogeneity of temporary employment jobs, in Chile it is more likely that the explanation lies in the fact that many permanent jobs are precarious. Based on data from this same survey, we have shown that the sensibility of temporary employment as indicator of precarious employment is low [22]. The implication is that temporary employment, as indicator of precarious employment, introduces non-differential exposure misclassification, producing an underestimation of the associations between precarious employment and health. This should be especially so in countries and contexts where most forms of employment tend to be precarious to some extent, as is the case of Chile. This does not preclude, however, the existence of associations between temporary employment and other health outcomes not included in this study, or for these same health outcomes in other contexts where the gap between temporary and permanent employment is greater [9].

Consistent with the literature, job insecurity exhibited a strong association with poor mental health, and, somewhat weaker, with poor general health. It did not, however, appear associated with occupational injuries, despite the expectations are that job insecurity affects the occupational health and safety of workers through different mechanisms such as over-exertion and performance pressures in order to preserve the job when it is perceived as insecure [23].

The strongest and most consistent pattern of associations with all three outcome variables were observed for multidimensional employment precariousness. This is consistent with our hypothesis that this is a more sensible measure and with greater explicative power than either job insecurity or temporary employment. Particularly interesting is the result of a strong, graded association between the EPRES and occupational injuries, association that was not observed for any of the other exposure variables. This is the first evidence of an association between the EPRES and occupational injuries, comparable to previous research on temporary employment [24–26], and consistent with the GREDS-EMCONET conceptual model on employment conditions and health, providing evidence to the proposal that precarious workers face worse working conditions and poorer occupational health and safety protection [27].

Also, given these results, self-self bias seems less likely given the higher objectivity of reporting occupational injuries as compared to general and mental health.

Results were also generally consistent with the expectation that groups in greater disadvantage are more vulnerable to poor employment conditions, possibly given cumulative vulnerability or the combination of several adverse exposures simultaneously or along the life course. We identified patterns of effect measure modification by sex, where employment precariousness appears to affect women's mental and general health more, which is consistent with previous studies [13, 14], and where job insecurity and temporary employment appear to affect men's mental health more than women's, suggesting that job instability is especially crucial for them and that other dimensions of employment precariousness may be more relevant to women (e.g., vulnerability or defencelessness, incapacity to exercise rights).

With one exception, all observed associations were modified by age, indicating a greater risk as workers' age increases, consistent with studies showing older workers are more likely to experience adverse effects in the face of insecure or precarious employment [24]. Heterogeneity by occupational class showed a clear pattern of lower vulnerability for those in non-manual occupations, especially concerning general health, the latter possibly due to a greater physical workload and a lower investment in manual workers' occupational health and safety, resulting in a greater wearing-off of their health. It is noteworthy that the employment precariousness scale better captured these group differences, with a larger number of associations exhibiting heterogeneity for this exposure than for the others.

This study is not without limitations. To begin with, it is cross-sectional in nature, and thus, exposed to possible issues of reverse causality or selection bias. Yet our results remained unchanged after excluding individuals reporting a limiting illness in the 12 months preceding the survey.

Another limitation is the exclusion of the most vulnerable workers in the Chilean labour force, given its focus on salaried workers with a formal job contract. If currently unemployed workers, own-account workers, or informal workers were included in such a study, we might expect even larger associations. However, including them requires adapted versions of the EPRES questionnaire that need yet to be developed. Future research should consider these knowledge gaps, as well as expanding the study to other health outcomes, e.g. occupational illnesses, and in other time periods, to explore what the effects of economic crises may be on the observed associations.

## Conclusions

In summary, our study shows that precarious employment conditions are harmful for health, but that different approaches to the measurement of precarious employment produce different results. According to the results presented here, the multidimensional employment precariousness scale is more sensible to the potential health effects of precarious employment than one-dimensional methods that only address employment instability, and is also more sensible to differences across groups of workers in their vulnerability to precariousness of employment. Another advantage of using a theory-based multidimensional measure is that it allows for the study of the health related effects along a gradient of precariousness irrespective of contract type. These results highlight the value of a multidimensional tool, for research as well as monitoring, in order to strengthen the evidence base for policy making to the benefit of workers' health.

## Supporting information

**S1 Table. Interaction terms (95% C.I.) for the associations between study exposures and outcomes by sex, age, education and occupation.**
(DOCX)

## Author Contributions

**Conceptualization:** Alejandra Vives, Tarik Benmarhnia, Joan Benach.

**Formal analysis:** Tarik Benmarhnia, Francisca González.

**Funding acquisition:** Alejandra Vives.

**Methodology:** Alejandra Vives, Tarik Benmarhnia.

**Project administration:** Alejandra Vives, Francisca González.

**Writing – original draft:** Alejandra Vives, Tarik Benmarhnia.

**Writing – review & editing:** Alejandra Vives, Tarik Benmarhnia, Francisca González, Joan Benach.

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
