## [Decision Letter · Decision Letter 0]

15 Jun 2020

PONE-D-20-09255

The importance of using a multi-dimensional scale to capture the various impacts of precarious employment on health: Results from a national survey of Chilean workers

PLOS ONE

Dear Dr. Vives,

Thank you for submitting your manuscript to PLOS ONE. After careful consideration, we feel that it has merit but does not fully meet PLOS ONE’s publication criteria as it currently stands. Therefore, we invite you to submit a revised version of the manuscript that addresses the points raised during the review process.

We look forward to receiving your revised manuscript.

Kind regards,

Semih Tumen, PhD

Academic Editor

PLOS ONE

Journal Requirements:

Reviewers' comments:

Reviewer's Responses to Questions

**Comments to the Author**

1. Is the manuscript technically sound, and do the data support the conclusions?

Reviewer #1: Partly

Reviewer #2: Partly

2. Has the statistical analysis been performed appropriately and rigorously? 

Reviewer #1: Yes

Reviewer #2: Yes

3. Have the authors made all data underlying the findings in their manuscript fully available?

Reviewer #1: Yes

Reviewer #2: Yes

4. Is the manuscript presented in an intelligible fashion and written in standard English?

Reviewer #1: Yes

Reviewer #2: Yes

5. Review Comments to the Author

Reviewer #1: Dear authors,

The manuscript develops and important, and still neglected, determinant of health. In the case of Chile and other similar countries, relationship among work and health is still full of stigma and a lack of comprehensive polices to undermine the negative effects of (Precarious) work in population health.

My suggestions are focused in the introduction, methods, and discussions. In the case of introduction section, there are no references supporting Chile´s relationship between precarious work and health. In the same way, current (/Or not) policies related to the problem. Thus, in methods is necessary to support why Chile -and not another country with similar problems- was selected for using a multi-dimensional scale for precarious work and health. If the main reason for choosing Chile was data, it must by explicitly declared in this section. Finally, discussion needs to avoid repeating results and discuss with findings. I am aware multi-dimensional scale for measure precarious work and health cannot be available in similar context.

Reviewer #2: The paper titled “The importance of using a multi-dimensional scale to capture the various impacts of precarious employment on health: Results from a national survey of Chilean worker” measures precarious employment by using one- and multi-dimensional indicators and assessing their relationship with several health outcomes. The paper seeks to contribute to the literature on precarious employment which, as they argue, has not really developed a clear and comprehensive operationalization of the concept.

The paper presents original research with results that, to my knowledge, have not been published elsewhere. However, there are certain issues with the paper that I think could be improved to make this paper a stronger contribution.

First of all, it is not fully clear the method the authors use. They state they use “multilevel modified poisson” models, where individuals are nested within regions. However, to my understanding, the data the authors are using is not representative at the regional level, therefore, the variances estimated at the regional level by a hierarchical model would not be valid. In that sense, I think it would be better to use a standard poisson model using clustered standard errors to adjust for the autocorrelation that might exist.

Regarding the measurement of their variables, it should be made clear that classifying individuals in the 4th quartile of the GHQ distribution as individuals with poor mental health is the standard in the literature. To what extent is that correlated with, for example, clinical depression. I imagine that adding sources that justify the way this is measure will not be hard and it will give the reader a higher sense of robustness in terms of measurement.

When it comes to the presentation of results, the use of tertiles for the employment precariousness variable is confusing. It is not mentioned before, until one reaches the table, that the variables is going to be used in tertiles rather than as a continuous variable. Why is the variable categorized? Is there a theoretical or methodoligcal justification? That should be made clear. Also, the analysis of “heterogeneity across sub groups” of the associations between precarious employment and health seems undone. At the very end of the results section that extra paragraph seems a little like added last minute and it is very briefly really discussed afterwards. In other words, it looks like a secondary objective that is not really adding to the paper. For me, the primary results are more than enough to make this a valuable contribution.

Finally, I cannot finish without commenting the results on temporary employment. Chile has two particular features that I think are very important to be discussed in the paper so that the results are actually valid. First, a very significant proportion of the labor force, as in the rest of Latin America, is in the informal sector. Because they have been left out of the analysis, these analyses are not really showing the whole pctire of precariousness. And don’t get me wrong, I think the analyses are not less valid because of this, but this should be made clear given the context in which the workers being analyzed live. It is very different to be a precarious worker in your sample---who is relatively more stable than a significant proportion of the labor force in their country---than a precarious worker in a country where most of the labor force are formally employed. And second, regarding the little effect of temporary jobs, it is important to mention that the Chilean law regulates temporarily employment, and in many cases, workers who are under a temporary contract are expected to move to long-term contracts in a relatively short period of time. Of course, this is not all workers, but it’s a significant part of them (maybe you can explore this with your data). But it could be that this fact is in part explaining part of that null finding.

Overall, I think this is a good paper, that makes a valuable contribution, and that needs to adjust some things in its methodology and presentation of results to make it an even stronger one. The article is presented in an intelligible fashion and is written in standard English, and there are no ethical concerns about the data being used. The article adheres to appropriate reporting guidelines and community standards for data availability.

6. PLOS authors have the option to publish the peer review history of their article (what does this mean?). If published, this will include your full peer review and any attached files.

Reviewer #1: No

Reviewer #2: No

---

## [Author Response · Author response to Decision Letter 0]

13 Aug 2020

Dear editor,

Please find below the reviewer comments, followed by our responses.

Editor

We note that you have included the phrase “data not shown” in your manuscript. Unfortunately, this does not meet our data sharing requirements. PLOS does not permit references to inaccessible data. We require that authors provide all relevant data within the paper, Supporting Information files, or in an acceptable, public repository. Please add a citation to support this phrase or upload the data that corresponds with these findings to a stable repository (such as Figshare or Dryad) and provide and URLs, DOIs, or accession numbers that may be used to access these data. Or, if the data are not a core part of the research being presented in your study, we ask that you remove the phrase that refers to these data.

We have excluded the phrase in line 265 and added the corresponding reference instead. It is a recently accepted manuscript in Annals of Work Exposures and Health. We have entirely excluded the second phrase relative to this ( lines 296-300 of the manuscript with track changes) since those dara were not a core part of the research being presented here.

Reviewer #1

The manuscript develops and important, and still neglected, determinant of health. In the case of Chile and other similar countries, relationship among work and health is still full of stigma and a lack of comprehensive polices to undermine the negative effects of (Precarious) work in population health.

We thank reviewer 1 for his positive appreciation of our manuscript. 

My suggestions are focused in the introduction, methods, and discussions.

1. In the case of introduction section, there are no references supporting Chile´s relationship between precarious work and health. In the same way, current (/Or not) policies related to the problem. 

We thank the reviewer for this observation. There is scant epidemiological literature supporting this association in Latin America in general, and in Chile in particular. We have made reference to this in the introduction section and provided reference to descriptions of both the labor market and policy at the time the data were collected. Also, to the preliminary descriptive results of the survey. Because the primary focus of this study is the comparison between indicators of employment precariousness, we do not extend ourselves more in that regard. 

2. Thus, in methods is necessary to support why Chile -and not another country with similar problems- was selected for using a multi-dimensional scale for precarious work and health. If the main reason for choosing Chile was data, it must by explicitly declared in this section. 

We agree with the reviewer and have added a statement in the introduction regarding both the pertinence of the analyses in the Chilean context, and the availability of data that justify the employment of the ENETS survey. 

3. Finally, discussion needs to avoid repeating results and discuss with findings. I am aware multi-dimensional scale for measure precarious work and health cannot be available in similar context.

We truly thank the reviewer for this observation. In fact, there are scant experiences with multi-dimensional scales for measuring precarious employment in the literature and more so regarding epidemiological research. To our knowledge, there is currently some research going on in central America, using an incomplete version of the EPRES questionnaire, but results have not been published. We have gone through the discussion to make it less repetitive and more synthetic and truly believe it has improved. 

Reviewer #2

The paper titled “The importance of using a multi-dimensional scale to capture the various impacts of precarious employment on health: Results from a national survey of Chilean worker” measures precarious employment by using one- and multi-dimensional indicators and assessing their relationship with several health outcomes. The paper seeks to contribute to the literature on precarious employment which, as they argue, has not really developed a clear and comprehensive operationalization of the concept. The paper presents original research with results that, to my knowledge, have not been published elsewhere. 

However, there are certain issues with the paper that I think could be improved to make this paper a stronger contribution.

1. First of all, it is not fully clear the method the authors use. They state they use “multilevel modified poisson” models, where individuals are nested within regions. However, to my understanding, the data the authors are using is not representative at the regional level, therefore, the variances estimated at the regional level by a hierarchical model would not be valid. In that sense, I think it would be better to use a standard poisson model using clustered standard errors to adjust for the autocorrelation that might exist.

Thanks for this comment. We would like to clarify that the data are representative at the regional level. We clarified this in the methods section We actually considered both the hierarchical structure of the data (by using fixed effects at the regional level with clustered standard errors) and also included the survey weights. We used modified Poisson models as our outcomes are binary following the approach proposed by Barros et al. [See ref 18: https://doi.org/10.1186/1471-2288-3-21]. Therefore, our data consider any time-fixed confounding at the regional level and potential autocorrelation within regions and our estimates are representative of the target working Chilean population. We clarified this in the abstract and the methods section. 

2. Regarding the measurement of their variables, it should be made clear that classifying individuals in the 4th quartile of the GHQ distribution as individuals with poor mental health is the standard in the literature. To what extent is that correlated with, for example, clinical depression. I imagine that adding sources that justify the way this is measure will not be hard and it will give the reader a higher sense of robustness in terms of measurement.

We thank the reviewer for this suggestion. In fact, the GHQ can be scored in different manners, this being the dimensional approach, which uses a Likert scoring method, and is frequently used in epidemiological research to assess intensity of psychological distress at the population level. It does not aim at a clinical diagnosis of depression nor as a clinical screening tool, for which the GHQ scoring approach is best recommended. With this method, a score is obtained which is a continuous variable, where higher scores indicate higher psychological distress. We have included a reference to support this utilization. Further, we changed “dimensional” for “Likert” in the manuscript to be in consistence with the reference. Here we dichotomize the GHQ score. This is a frequently used strategy, and the choice here, in order to produce a dichotomous variable for the proposed analytical strategy.

3. When it comes to the presentation of results, the use of tertiles for the employment precariousness variable is confusing. It is not mentioned before, until one reaches the table, that the variables is going to be used in tertiles rather than as a continuous variable. Why is the variable categorized? Is there a theoretical or methodoligcal justification? That should be made clear. 

We thank the reviewer for this observation. The employment precariousness scale is scored as a continuous variable and has been most frequently been used in quantiles in order to show non-linear associations with the outcome. In this study, we a priori used tertiles for employment precariousness to investigate potential non-linear patterns while keeping enough observations in each group. We agree this information should be clarified further in the text, so included a statement about this in the methods section (lines 197-199). 

4. Also, the analysis of “heterogeneity across sub groups” of the associations between precarious employment and health seems undone. At the very end of the results section that extra paragraph seems a little like added last minute and it is very briefly really discussed afterwards. In other words, it looks like a secondary objective that is not really adding to the paper. For me, the primary results are more than enough to make this a valuable contribution.

We totally agree with the reviewer. Indeed, this effect measure analyses do not constitute a primary aim but rather an exploratory/secondary aim and we should have clarified this. In the revised version, we clarified that such analysis constitutes a secondary aim and we moved the results table to the appendix section. Consistent with this, we also reduced the length of the corresponding paragraph in the results and discussion sections. 

5. Finally, I cannot finish without commenting the results on temporary employment. Chile has two particular features that I think are very important to be discussed in the paper so that the results are actually valid. First, a very significant proportion of the labor force, as in the rest of Latin America, is in the informal sector. Because they have been left out of the analysis, these analyses are not really showing the whole picture of precariousness. And don’t get me wrong, I think the analyses are not less valid because of this, but this should be made clear given the context in which the workers being analyzed live. It is very different to be a precarious worker in your sample---who is relatively more stable than a significant proportion of the labor force in their country---than a precarious worker in a country where most of the labor force are formally employed. 

We entirely agree with the reviewer. In the limitations section of the discussion we refer to this exclusion of the most vulnerable workers, that is, informal workers. As we indicate there, their inclusion requires an adaptation of the employment precariousness scale, and also a different conceptualization if we were to include self-employed workers, an important part of our informal labor force. We aim here to reinforce the notion that multidimensional measures of employment precariousness are needed to adequately assess its impact on health, and to discuss with the literature concerning the limitations of other approaches. However, we do agree with the reviewer that this is a critically important topic, and which requires careful formulation of concepts and measurement instruments to address these groups. We hope the reviewer will find our discussion of these limitations in the paper satisfactory. 

6. And second, regarding the little effect of temporary jobs, it is important to mention that the Chilean law regulates temporarily employment, and in many cases, workers who are under a temporary contract are expected to move to long-term contracts in a relatively short period of time. Of course, this is not all workers, but it’s a significant part of them (maybe you can explore this with your data). But it could be that this fact is in part explaining part of that null finding.

We thank the reviewer for this observation. This is an interesting hypothesis, although we cannot directly explore in the data. We have, however, analyzed the extent of precariousness of temporary workers in another paper (now cited in the manuscript) and have found that they are indeed almost all in precarious job situations and are more intensely precarious than jobs with permanent contracts. The issue is, however, that many precarious jobs are permanent jobs, so the use of this indicator relying on type of contract produces misclassification error and thus underestimates the associations. We have indicated this more clearly in lines 263-268.

Overall, I think this is a good paper, that makes a valuable contribution, and that needs to adjust some things in its methodology and presentation of results to make it an even stronger one. The article is presented in an intelligible fashion and is written in standard English, and there are no ethical concerns about the data being used. The article adheres to appropriate reporting guidelines and community standards for data availability.

We thank the reviewer for this positive evaluation of our manuscript.

---

## [Editor Report · Decision Letter 1]

17 Aug 2020

The importance of using a multi-dimensional scale to capture the various impacts of precarious employment on health: Results from a national survey of Chilean workers

PONE-D-20-09255R1

Dear Dr. Vives,

We’re pleased to inform you that your manuscript has been judged scientifically suitable for publication and will be formally accepted for publication once it meets all outstanding technical requirements.

Kind regards,

Semih Tumen, PhD

Academic Editor

PLOS ONE

---

## [Editor Report · Acceptance letter]

15 Sep 2020

PONE-D-20-09255R1 

The importance of using a multi-dimensional scale to capture the various impacts of precarious employment on health: Results from a national survey of Chilean workers 

Dear Dr. Vives:

I'm pleased to inform you that your manuscript has been deemed suitable for publication in PLOS ONE. Congratulations! Your manuscript is now with our production department. 

Kind regards, 

on behalf of

Professor Semih Tumen 

Academic Editor

PLOS ONE